Exploring of cardiac autonomic activity with heart rate variability in long-term kratom (Mitragyna speciosa Korth.) users: a preliminary study

Saengmolee Wanumaidah 1 2 3
Cheaha Dania 1 4
Sa-ih Nusaib 1 2
Kumarnsit Ekkasit 1 2 kumarnsit.e@gmail.com
1 Biosignal Research Center for Health, Faculty of Science, Prince of Songkla University , Hatyai, Songkhla , Thailand
2 Physiological Program, Division of Health and Applied Sciences, Faculty of Science, Prince of Songkla University , Hatyai, Songkhla , Thailand
3 Bio-Inspired Robotics and Neural Engineering (BRAIN) Laboratory, School of Information Science and Technology (IST), Vidyasirimedhi Institute of Science and Technology (VISTEC) , Rayong , Thailand
4 Biology Program, Division of Biological Science, Faculty of Science, Prince of Songkla University , Hatyai, Songkhla , Thailand
Pazzaglia Mariella
Electronic publication date: 2022 Oct 25
Publication date: 2022
Volume: 10
Electronic Location ID: e14280
Received 2022 Feb 24; Accepted 2022 Sep 30
Copyright: © 2022 Saengmolee et al.
Copyright year: 2022
Copyright holder: Saengmolee et al.
License: This is an open access article distributed under the terms of the Creative Commons Attribution License, which permits unrestricted use, distribution, reproduction and adaptation in any medium and for any purpose provided that it is properly attributed. For attribution, the original author(s), title, publication source (PeerJ) and either DOI or URL of the article must be cited.
License URL: https://creativecommons.org/licenses/by/4.0/

Keywords: Long-term kratom consumption, Cardiac autonomic function, Heart rate variability, Frequency domain analysis

Funding: Program of Physiology, Faculty of Science Graduate School Dissertation Funding for Thesis Revenue Budget Fund from Prince of Songkla University Educational Institutions Scholarship for Outstanding GPA The present study was supported by the Program of Physiology, Faculty of Science, grants funded by the Graduate School Dissertation Funding for Thesis and Revenue Budget Fund from Prince of Songkla University and Educational institutions Scholarship for Outstanding GPA. The funders had no role in study design, data collection and analysis, decision to publish, or preparation of the manuscript.

==============================
Background

Kratom is a psychoactive plant used to enhance productivity among laborers in Southeast Asian countries. Previous findings from in vitro research of mitragynine, a major component of kratom, suggested a possible risk of heart function abnormality. However, the cardiac autonomic function in long-term kratom users with chewing forms has never been studied. This study aimed to investigate heart rate variability (HRV) indices of cardiac autonomic function in long-term kratom chewers (LKC), compared to the control levels, and also to examine the correlation between HRV indices and relevant kratom use factors.

Method

A total number of 50 participants consisted of LKC (n = 31) who regularly chewed fresh kratom leaves for at least 2 years and demographically matched control subjects (n = 19). Resting electrocardiogram (ECG) signals were recorded from subjects for 3 min to analyze the ultrashort HRV in the frequency domain. The normalized low frequency (LFn) and high frequency (HFn) were chosen to be the HRV indices to evaluate cardiac autonomic function. The comparison of HRV indices between groups and the correlation between HRV indices and duration and quantity of kratom use was further conducted in statistical analysis.

Results

The LKC significantly increased LFn together with enhanced HFn compared to the control group tested, indicating that LKC changed cardiac autonomic function with parasympathetic dominance. Furthermore, no significant correlation between the HRV indices and the duration and quantity of kratom use was found, suggesting that the HRV indices were not relevant to these factors. The present study provided scientific-based evidence of cardiac autonomic modulation in long-term kratom chewers. LFn and HFn may be promising cardiac autonomic indicators for monitoring health outcomes in LKC.

Introduction

Kratom or Mitragyna speciosa is a traditional herbal medicine in Southeast Asia, particularly in rural areas of Thailand and Malaysia (Hassan et al., 2013). Due to its stimulant and opioid-like effects, it was placed under schedule 5 of the Thai Narcotics Act, making it illegal to sell or possess kratom in the past. Nowadays, the Thai government has delisted the kratom plant as a narcotic to be partially legalized in the context of traditional and medical purposes (Charoenratana, Anukul & Aramrattana, 2021). However, the scientific evidence of long-term kratom consumption associated health outcome in human remains elusive, particularly in cardiac function.

Kratom typically promotes the dual effect depending on the dosage of use. The low dose exerted a stimulant effect while the high dose presented an opioid-like effect (Prozialeck, Jivan & Andurkar, 2012). According to pharmacological profiles, mitragynine and 7-hydroxymitragynin are active components in kratom leaves extract (Veltri & Grundmann, 2019). Both of them can act on opioid and non-opioid receptors. The later receptors, such as alpha-2 receptor, had effects on heart rate (Ellis et al., 2020). The effect of mitragynine on cardiac function were investigated both in vitro and in human study. The molecular evidence of cardiotoxicity from in vitro experiment was one of the major concerns about kratom effects. Kratom was found to suppress potassium current (IKr) that might cause long QT syndrome, resulting in lethal cardiac arrythmias (Tay et al., 2019). However, regular kratom users who consumed kratom tea containing 434.28 mg of mitragynine on a daily did not appear to have electrocardiogram (ECG) abnormalities with prolonged QT, except for a higher risk of sinus tachycardia with a higher average heart rate measured at a resting baseline compared to non-kratom using controls (Leong Abdullah et al., 2020). Regarding the evidence, regular users who drink kratom tea appear to show sympathetic dominance.

In contrast to kratom tea consumption, the different form of long-term kratom consumption was also found in older people who chewed kratom leaves in the traditional style (Charoenratana, Anukul & Aramrattana, 2021). They usually chewed kratom leaves at very high amounts (10–30 kratom leaves per day) to prolong and tolerate extreme fatigue (Henningfield, Fant & Wang, 2018). With high quantities of consumption, they experienced relaxation and calm (Johnson et al., 2020). It is possible that those LKC may have a predominant parasympathetic activity. However, these effects of kratom consumption have never been directly confirmed with measurable autonomic cardiac activity.

Heart rate variability (HRV) is a noninvasive and inexpensive method to detect the autonomic cardiac function modulated by heart-brain interaction. It describes the oscillatory patterns of time intervals between consecutive adjacent heart beats (electrocardiogram RR intervals) (Shaffer & Ginsberg, 2017). High HRV exhibited in healthy subjects was found to reflect the regulation of the ANS in adaptation to environmental and psychological challenges, whereas low HRV was found in psychological and pathological problems (Maule et al., 2008; Henry, Minassian & Perry, 2012; Ingjaldsson, Laberg & Thayer, 2003; Barutcu et al., 2005).

In general, the gold standard of HRV measurement required 24-h and 5-min periods for long- and short-term ECG recording, respectively. The short-term HRV is more widely used than the long-term HRV due to its practical process and data analysis. However, the standard HRV evaluation is still long for routine healthcare purposes that might cause discomfort during recording (Kim, Seok & Shin, 2021). In our preliminary study, local people found the ECG recording uncomfortable and difficult to withstand. Therefore, an ultra-short HRV (less than a 5-min ECG recording) is required to reduce possible discomfort during signal recording. In the present study, the 3-min ultra-short HRV was chosen.

Two HRV indices in frequency domain analysis: low frequency (LF) and high frequency, were usually used for ultra-short HRV analysis. HF component required a minimum recording for 1 min while LF component required at least 2 min (Malik et al., 1996). Values of the HF component typically reflect parasympathetic activity (Shaffer & Ginsberg, 2017), while the LF component is a more controversial interpretation. Some reports defined it as a sympathetic activity, while others indicated the marker of both sympathetic and parasympathetic modulations. To clearly interpret the index of sympathetic activity, the LF values needed to be normalized by the total power. Since the total power is changed in the direction of LF and HF when spectral components are computed in an absolute unit (milliseconds squared) (Papaioannou et al., 2007). Moreover, the frequency components in the absolute unit produced the power fluctuation. Therefore, the normalized LF (LFn) and HF (HFn) values are used to attenuate the variation of power distribution (Vishwajeet, Singh & Deepak, 2020).

The present study aimed to determine cardiac autonomic function assessed by ultra-short HRV with LFn and HFn indices in LKC in comparison to non-kratom using controls. Moreover, the relationship between HRV indices and kratom use variables, such as duration and quantity of kratom use, was also investigated. The LKC group was hypothesized to show a significant increase in HRV by enhancing the HFn and reducing the LFn indices of cardiac autonomic function at rest, compared to the control levels. Moreover, the LFn and HFn would be positively correlated with the kratom use variable.

Materials and Methods

Study design and participants

A final sample of 50 male subjects, consisting of LKC (n = 31) and non-kratom using controls (n = 19) were recruited from two local areas in southern Thailand (Bannasarn district, Surat Thani province and Natawee district, Songkhla province). In these regions, kratom plants have been widely used in cultural activities and daily life. The present study was conducted by following the Declaration of Helsinki and the protocol was approved in writing by the Ethics Committee of Prince of Songkla University, Thailand (HSc-HREC-63-017-1-1). The following criteria were used to include kratom users: (a) age over 18 years; (b) daily kratom chewing (2 years); (c) last time chewing at least 2 h to avoid the peak of kratom’s acute effect (Singh et al., 2019); (d) no history of psychiatric diseases as measured by the Brief Psychiatric Rating Scale (BPRS); and (e) no history of illicit drug use and no underlying. In the cases of the control, they had the same background as found in long-term kratom chewers, except their confirmation that they did not use kratom at all. The LKC were recruited at first. After that, control subjects were recruited to match the background variables of LKC. All participants provided verbal informed consent to participate in the study. Finally, the participants were asked to complete the questionnaires to collect the socio-demographic characteristics such as age, sex, alcohol drinking, and kratom use history (e.g., duration or amount of kratom use per day).

ECG acquisition and HRV analysis

After participants completed the questionnaires, resting ECG signals were recorded during an eye-closed period for 3 min. Although the standard short-term HRV measurement was recommended for 5-min recordings (Shaffer & Ginsberg, 2017), the ultra-short term HRV features (less than 5 min) also showed good surrogate of HRV study (Castaldo et al., 2019). The measurement of ECG signals was acquired using a standard limb lead I configuration. The Ag/AgCl solid adhesive pre-gelled electrodes (Tian run, Beijing, China) were fixed on the left and right sides of the clavicle with positive and negative electrodes, respectively, with one grounded electrode attached to the participant’s right ankle. These positions were chosen to eliminate movement artifacts (Sarang & Telles, 2006). ECG signals were recorded using BioAmp and Powerlab systems (ADInstruments, Castle Hill, Australia) connected to LabChart software (version 7) at sampling rate of 1 k/s. The recorded ECG signals were band-pass filtered at 0.3–30 Hz according to previous studies (Baumert et al., 2007; Palma et al., 2013, 2015).

The HRV analysis was performed by NeuroKit2: a Python toolbox for neurophysiological signal processing (Makowski et al., 2021). Briefly, raw ECG signals were then transferred to a laptop computer for storage and offline analysis. The ECG signals were first cleaned by using ecg_clean (). This function was also applied for artifact correction of the RR-interval series by inputting the argument of artifact_correction. This argument was based on the artifact detection algorithm as defined by the previous study (Lipponen & Tarvainen, 2019). Then, HRV in the frequency domain was computed by Welch method with the hrv_frequency () function to extract the distribution of power spectrum density across the low-frequency (LF; 0.04–0.15 Hz) and high-frequency (HF; 0.15–0.4 Hz). The normalized low-frequency (LFn) and high frequency (HFn) components that are relative to the total power were used as HRV indices in the present study. The LFn index represented sympathetic influence while the HFn index represented parasympathetic function in the cardiac activity.

Statistical analysis

The socio-demographic data were compared between two groups using chi-square test for proportion difference and independent t-tests for the difference between groups.

The HRV indices (LFn and HFn) were first tested for their normal distribution before conducting a parametric test. The data by groups did not show statistical significance by Kolmogorov-Smirnov test indicating that data exhibited normal distribution. The present data were unequal sample sizes assuming unequal variances. A Welch’s t-test for unequal variances (Delacre, Lakens & Leys, 2017) was conducted to compare means of HRV indices between the control and LKC groups. The effect size from our preliminary was tested to evaluate the magnitude difference between groups according to Cohen’s guidelines (Cohen, 1988), where d = |0.2| is considered as small, d = |0.5| as medium, and d = |0.8| as large effect size. A power analysis was also conducted based on the effect size. The Welch’s t-test was carried out using R studio (version 4.1.2).

To test whether HRV indices were associated with kratom use variables (duration and the quantity of kratom leaves use per day), Spearman’s rank correlation coefficient was performed as some kratom variables showed a non-normal distribution. Since internal factor such as age was relevant to the kratom use variables and HRV indices (Abhishekh et al., 2013), it was included in the correlation test to determine the relationships among of them. Spearman’s rank correlation was tested and plotted by Python SciPy library.

All statistical significance levels were considered at p < 0.05.

Results

Characteristics of the participants and kratom use history

The socio-demographic characteristics between control and LKC are shown (Table 1). There was no significant difference in age, BMI, smoking, and alcohol drinking history between control and LKC. This means that the backgrounds did not differ between groups, indicating that participants’ backgrounds were not significantly different between groups. The average kratom chewing duration of LKC was 18.13 ± 11.04 years (mean ± SD) with the average amount of consumption at 11.65 ± 10.12 leaves/day (mean ± SD).

Table 1 The characteristics of participants.

Demographic characteristic	Control group	LKC group	p-value	
Number of subjects, n (%)	19 (38)	31 (62)	–	
Sex				
Male	19 (100)	31 (100)		
Female	–	–		
Age in years, mean ± SD	48.84 ± 10.66	51.32 ± 10.66	0.382a	
Body-mass index (BMI) in kg/m2,
mean ± SD	23.87 ± 2.82	23.03 ± 4.06	0.430a	
Smoking history			0.118b	
Non-smoking, n (%)	9 (47.30)	8 (25.80)		
Smoking, n (%)	10 (52.70)	23 (74.20)		
Alcohol drinking			0.640b	
Non-alcohol drinking, n (%)	8 (42.10)	11 (35.48)		
Alcohol drinking, n (%)	11 (57.90)	20 (64.52)		
Duration of kratom use, mean ± SD	–	18.13 ± 11.04	–	
The amount of kratom consumption (leaves/day), mean ± SD	–	11.65 ± 10.12	–	
Notes:

a Independent t-test.

b Chi-square test.

Ultra-short HRV in frequency domains between groups

The ultra-short HRV in spectral distributions across frequency components between the representative subjects in control and LKC groups was illustrated (Fig. 1). Compared to control levels, LKC appeared to have a lower LFn distribution, whereas an HFn distribution was higher. In quantitative analysis, the Welch’s t-test was conducted to compare the mean of the indices between groups, as shown (Fig. 2). When compared to control levels, LFn in LKC was significantly decreased [t = 4.5842, df = 43.819, p < 0.001, d = 1.21 (large effect), observed power = 0.98], whereas HFn was significantly increased [t = −4.5193, df = 45.619, p < 0.001, d = 1.21 (large effect), observed power = 0.98]. The results suggested that the long-term kratom chewer group exhibited the change in cardiac autonomic function with parasympathetic predominant observed by the increase in HFn and decrease in LFn.

Figure 1 The effect of long-term kratom chewing on the HRV in frequency domain analysis.

The distribution of power spectrum in normalized values across the low-frequency (LF; 0.04–0.15 Hz) and high-frequency (HF; 0.15–0.4 Hz) for the representative subjects of controls and LKC.

Figure 2 The statistical comparisons between groups’ differences for HRV in frequency domain analysis.

The comparisons between groups’ differences (Control n = 19, LKC n = 31) for the LFn and HFn were tested by Welch’s t-test. Asterisks (***) denote p < 0.001.

The correlations between changes in HRV indices in LKU and the variables of kratom use

The significant differences in LFn and HFn indices in LKC compared to the controls were further investigated in association with the relevant factors, such as duration, the quantity of kratom leaves use, and age. The duration, except for the quantity of kratom use tended to correlate to LFn positively and HFn negatively. The tendency of the correlation may be a result of age. Since age was likely to correlate with the HRV indices as well. Moreover, age was highly associated with the duration rather than the quantity of kratom use. This could explain why only the duration of kratom use was associated with a higher tendency with those HRV indices. However, all HRV indices were not significantly correlated with these factors tested by spearman’s correlation (Fig. 3). The results suggested that changes in the HRV indices in LKC were independent of kratom use variables.

Figure 3 The correlation between HRV indices in LKC (n = 31) and the variables of kratom use, including age.

The pairwise scatter plots are represented in the left corner, while Spearman’s correlation coefficients are illustrated in the right corner with the different colors and sizes of the circle shapes.

Discussion

This preliminary study first evaluated cardiac autonomic function using HRV analysis in the LKC from Southern Thailand compared to that of kratom non-users. Recently, the evidence of autonomic heart regulation in the LKC remained unexplored. Investigation of HRV was important for a more comprehensive understanding of long-term kratom chewing on the dynamics of cardiac activity. The current findings supported some of the hypotheses that LKC showed some changes in cardiac autonomic function with increased HRV by enhancing HFn and reducing LFn during the resting state compared to control levels. However, the alterations in the HRV indices were not associated with the duration and quantity of kratom use.

The present results revealed that long-term kratom chewing increased HRV (increase in HFn and decrease in LFn). A previous study reported that the reduction of sympathetic tone had a similar effect to the increase in parasympathetic reactivity (Levin et al., 2019). The present data demonstrated that LKC changed in cardiac autonomic function that produced parasympathetic predominance. However, the results were inconsistent with the report from regular kratom users who daily consumed brewed kratom tea that produced sinus tachycardia, indicating increased sympathetic activity (Leong Abdullah et al., 2020). This could be a result of different forms of Kratom consumption. The boiling preparation contained diluted kratom content and lower bioactivity in comparison to chewing (Eaimchaloay, Kalayasiri & Prechawit, 2019).

Considering the form of kratom chewing, the increased HRV representing the predominant parasympathetic activity in the present study may be supported by the findings of physiological measurements such as EEG activity. Previously, the heart-brain interaction did exist (Jung, Jang & Lee, 2019). Basically, both sympathetic and parasympathetic activities of the autonomic nervous system are regulated by cortical-subcortical pathways (Thayer et al., 2009). A previous study revealed that long-term kratom chewers promoted an increased theta/alpha ratio (increase in theta and decrease in alpha band) at resting-state (Saengmolee et al., 2022). In addition, the change in EEG oscillation was a common response at rest found in opioid users (Phillips, Heming & London, 1994; Wang, Kydd & Russell, 2016). It is well known that opioids enhance parasympathetic activity (Jiang et al., 2009). Based on the evidence, it is possible to confirm that LKC exerted autonomic cardiac modulation by producing parasympathetic predominance.

Although the long-term kratom chewers showed the dominant vagal tone observed by enhanced HRV indices (LFn and HFn), they were not significantly related to the duration and the amount of kratom leaves used. The results did not support the hypothesis that long-term kratom chewing would have a negative and positive correlation with LFn and HFn, respectively. The present data were not consistent with those of the previous study which showed that EEG activity was strongly associated with the amount of kratom leaves used (Saengmolee et al., 2022). The changes of HRV indices in LKC were not correlated with the amount of kratom use variables but found to be associated with cardiac autonomic modulation. It is likely that the cardiac autonomic function in LKC was modified in adaptation to the sympatho-vagal balance. In particular, the volunteers consumed kratom leaves in daily life for almost two decades. Previously, the cardiac autonomic function was shifted from vagal to sympathetic dominance through the heart regulation of peripheral receptors such as arterial baroreceptors (Parati, 2005).

The present study provided scientific evidence that had never been examined with the practical method for evaluation of cardiac autonomic function with rapidly quantitative measurements in long-term kratom chewers. In the light of the results, long-term kratom chewers were found to show the enhanced parasympathetic activity represented by increased HRV compared to control level. Basically, the higher HRV was considered as a healthy index normally observed in healthy individuals, athletes, and performing yoga (Papp et al., 2013). In further studies, it would be important to apply the HRV assessment for understanding how alterations in autonomic function correspond with other health outcomes such as emotional self-report and well-being and sleep quality in LKC.

A limitation might also be discussed in this study. The HRV was analyzed using raw ECG signals recorded in a relatively short time period. Basically, traditional short-term HRV recording recommended a minimum of 5 min (Min et al., 2008). However, the ultra-short-term HRV recording (less than 5 min) was practical and effective in analysis in the previous study (Lee et al., 2020). Moreover, female kratom users were also surveyed. However, no female respondent was recruited according to the culture and local attitude. They were reluctant to participate. Mainly, female kratom users chew kratom occasionally for traditional remedy (Assanangkornchai et al., 2007; Singh, Narayanan & Vicknasingam, 2016). The plasma concentration of mitragynine that would indicate more accurate amount of consumption was not measured in the present study. However, the averaged amount expressed in leaves per day was also useful for estimation. Further research should be conducted on HRV assessment along with plasma concentration measurement of mitragynine for a more comprehensive understanding of changes in cardiac autonomic function in LKC.

Conclusions

Our preliminary study demonstrated a change in cardiac autonomic function using LFn and HFn parameters to evaluate HRV in LKC. Moreover, the HRV indices in correlation with the duration and quantity of kratom leaves used were also examined. We found that LKC modulated cardiac autonomic function at rest by enhanced parasympathetic activity with a higher HFn, together with a lower LFn. However, the change in cardiac autonomic function was not correlated with the duration and amount of kratom leave consumption indicating that the autonomic regulation may be modified through the adaptation of sympathovagal balance. In further studies, The HFn and LFn parameters might be monitored as particular HRV indices of cardiac autonomic function to evaluate psychological outcomes of treatment in people with daily stress or anxiety. Moreover, these HRV indices are recommended to be used along with other physiological biosignals for additional understanding of correlation between measurable parameters of body response and emotional status.

Supplemental Information

Supplemental Information 1 Raw data.

The heart rate variability indices between control and long-term kratom user and the linear discriminant analysis for classification between-group.

Click here for additional data file.

Additional Information and Declarations

Competing Interests

Author Contributions

Field Study Permissions

Data Availability

The authors declare that they have no competing interests.

Wanumaidah Saengmolee conceived and designed the experiments, performed the experiments, analyzed the data, prepared figures and/or tables, authored or reviewed drafts of the article, and approved the final draft.

Dania Cheaha conceived and designed the experiments, performed the experiments, prepared figures and/or tables, and approved the final draft.

Nusaib Sa-ih performed the experiments, prepared figures and/or tables, and approved the final draft.

Ekkasit Kumarnsit conceived and designed the experiments, performed the experiments, authored or reviewed drafts of the article, and approved the final draft.

The following information was supplied relating to field study approvals (i.e., approving body and any reference numbers):

This study was approved by the Ethics Committee of Prince of Songkla University, Thailand (HSc-HREC-63-017-1-1).

The following information was supplied regarding data availability:

The raw data is available in the Supplemental File.

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
