# Peer review of "Exploring of cardiac autonomic activity with heart rate variability in long-term kratom (Mitragyna speciosa Korth.) users: a preliminary study"

_PeerJ, doi:10.7717/peerj.14280_

## Round 0.1 · original submission · Major Revisions

Thank you for submitting your manuscript to our journal.

As usual, I have invited comments from experts from your research domain.

As you will see, multiple limitations were highlighted by reviewers that would need to be addressed/explained carefully. There are a few more issues that deserve some of your attention, but the comments are very constructive and clear, and I think you will find it easy to take them on board. I would like to add, based on my own reading, that interpreting the findings requires some extra work to improve clarity.

Taken altogether, let me invite you to prepare a revision that addresses the issues, together with a cover letter explaining how you did so. My plan is to resend the revision to the present referees.

Reviewer 1 ·

Basic reporting

The English language used throughout is not as clear as it could be. I suggest getting a fluent English speaker to thoroughly edit each sentence, perhaps adding them on as an author. I am not able to provide suggestions for each grammar or language error, but here are some examples:
- A confusing sentence from the abstract: “Previously, mitragynine, its major component, was demonstrated in vitro that might suggest possible risk of heart function abnormality.” This could be changed to something like: “Previous findings from in vitro research of mitragynine, the major component of Kratom, suggest possible risk of heart function abnormality.”
- From this sentence in the introduction, I am still not sure if Kratom is legal or not: “According to its stimulant and opioid-like effects, it was regulated under legal status. Recently, the Thai government has delisted kratom plant as a narcotic to be legalized in the context of traditional and medical purposes”
- From the introduction: “changes in the heart function” should be “changes in heart function”
- From the “Characteristics of the participants” Results subsection: “It means that the backgrounds did not differ between groups.” This sentence seems odd. Changing “It” to “This” would be an improvement, but it might be better to add a clause like this to the end of the previous sentence: “…, indicating that participants’ backgrounds did not significantly differ between groups.”
- From the Results section: “By the meantime, the combination of sympathetic and parasympathetic activity (nLF), and sympathovagal balance (LFn/HFn and SD1 /SD2) were significantly decreased and increased, respectively in LKU compared to control levels”. This sentence should be re-written, and you could replace “By the meantime” with something more professional like “Conversely”.
- From Discussion section: “Previously, changes in HRV are crucial for indicating the dynamics of cardiac activity”. This sentence could be improved to something like: ‘Changes in HRV parameters have been shown to be crucial for indicating the dynamics of cardiac activity and health outcomes’

Intro

Overall, The introduction currently lacks sufficient literature review and background information. Though the paper should be brief, reflecting the preliminary nature of the study, the introduction needs 1) concise explanation of the background literature, 2) description of the importance of the topic and the gap in the literature this study fills, and 3) clearly stated research question(s) that the current study seeks to answer.
- Much more information is needed to expand on this sentence: “The evidence of cardiotoxicity from in vitro experiment was one of major concerns of kratom effects (Tay et al., 2019)”. What was the evidence for cardiotoxicity? What is the likelihood this translates to an in vivo issue?
- More information needed on this sentence: “The abnormalities seen in electrocardiograms (ECG) with sinus tachycardia was reported in regular kratom users who daily consumed kratom leaves containing 434.28 mg total mitragynine”. Was this a baseline or task-based ecg measure? Was tachycardia seen in all subjects or was there a higher average HR for users versus non-users? How is this study different from the study the current paper is describing?
- The explanation of HRV confusing: “Basically, HRV was greater during resting state than while performing a cognitive task.” Here, “was” makes it sound like you’re reporting study results, but it’s not clear where this info comes from.
- More background information is needed on what each metric of HRV signifies and how that applies or correlates with health outcomes (or whatever the authors’ main interest is). It is not helpful to just report that HRV has been studied in many different populations: “In addition, HRV was applied in emotional assessments such as in the cases with depression (Dell’Acqua et al., 2020).”
- Supplemental table 1, is helpful to expand on what each parameter represents, but it is probably more important that you briefly describe the implications of each parameter here in the text instead of in the table. For instance, following the first half of this sentence, you could add a second sentence stating that HF indexes the parasympathetic influence on the heart, and LF, though previously thought to index sympathetic activity, is influcenced by both branch of the autonomic nervous system: (“Frequency domain analysis of the oscillation RR intervals in time period is separated into powers of different frequency bands that indicate sympathetic and parasympathetic activities”)
- You say in the supplemental table 1 that LF/HF indicates “Sympathovagal balance”, but this is not the current understanding (Billman, 2013), and it is not really necessary to report it since you already report LF and HF.
- It seems unnecessary to test the effects in time, frequency, AND non-linear domains. These are going to be highly interrelated. (In fact, Shaffer and Ginsberg (2017) state: “The RMSSD is identical to the non-linear metric SD1, which reflects short-term HRV (Ciccone et al., 2017).”). You should pick the metric(s) that will best answer your research question (“what are effects of long-term kratom chewing on cardiac autonomic regulation?”)

Methods section, Design and participants subsection
- Did the control subjects report never using Kratom at all, not chewing regularly/recently, or was there a mix?

ECG acquisition subsection
- Why was 3 minutes chosen when 5 is the standard convention (and only takes 2 more minutes)? The authors give rationale that shorter than 5 minutes is acceptable, but as the reader, I am not sure why the researchers choose to collect less data when they could have collected more.
- Why was a band-pass filter of 0.3 to 30 Hz used?

HRV analysis
- Was any visual inspection for artifact or artifact correction made?
- Please explain your reasoning for reporting and analyzing normalized units rather than absolute values for the frequency domain parameters
Statistical analysis
- Is the sample size large enough for MANCOVA?

LDA
- With the addition of the LDA, it seems the authors are doing far too much with too few data
- Why did the authors standardize the already log-transformed data before running the LDA?
- Explain what the abbreviations TP, TN, FP, and FN stand for

Results section, “HRV indices across all domains between groups” subsection

- “Compared to control levels, the distribution of the RR interval of LKU in time domain was not different”. This sentence is true, but possibly misleading – yes, the distribution for both groups is normal, but the center values are quite different. It also seems unnecessary to state this here. since this is already stated in the caption for Figure 2.
- The MANCOVA seems unnecessary, and given the sample size, I’m not sure this study was properly powered to do a multivariate analysis like this. Also, be more specific than this sentence: “…indicating an overall significant difference between controls and LKU”. Your data can only indicate differences on the measured cardiac indices, not overall differences.
“Classification with LDA to detect LKU”
- LDA results may be confounded by the overlapping information provided by many of the parameters used.
- Did the researchers pre-define what the cutoff would be for “successfully distinguishing”? Are 73% and 77% generally accepted in the literature? (“The classification results indicated that the optimal classifiers (RMSSD and LFn/HFn) successfully distinguished LKU from controls.”)
Discussion section
- The sentence that begins with “Basically, acute effect of kratom consumption is known to produce dual effects…” is an example of information that should be in the introduction.
- Another limitation is that the form of kratom consumption (e.g., boiling) was not asked of participants. This could have provided clues as to why dose and duration of use was not correlated with HRV.
- The hypothesis that is stated in the discussion (“…the hypothesis that long-term kratom consumption modulated autonomic regulation at rest with enhanced parasympathetic activity defined by parasympathetic parameters of HRV indices compared to controls”) does not match what was stated in the introduction (“…hypothesized that the cardiac autonomic activity assessed by HRV indices in long-term kratom users was different from controls.)

Figures
- Figure 1 is not really necessary. If you wanted to include it to show that part of the processing methods, it should be changed to supplementary, and more information should be given (e.g., the x-axis is just labelled “samples” – does that mean it’s an average for all participants? A sample of data provided by the software?)
- The T-waves in Figure 1 are very large, which calls into question whether data were correctly filtered
- Add the units (ms) in the x-axis label of Figure 2A

Experimental design

Why were only 19 control participants recruited when the LKU group n=31?

It is redundant to say that HF, LF AND the LF/HF ratio all changed. The addition of LF/HF provides no new information.

There is a multicollinearity issue with the LDA. You should either remove the LDA piece from all sections of the paper, or should go back to the drawing board and choose 2 or 3 parameters that would best answer your question and only analyze those.

Validity of the findings

I agree with the authors’ main finding that Kratom increased the dominance of the parasympathetic nervous system at rest, thereby altering sympathovagal balance. However, the data seems to be over-analyzed considering 1) the rather small sample size, 2) the unbalanced group sizes, and 3) the high covariance of the nine parameters used.

Additional comments

This paper represents an excellent preliminary study. I applaud the authors for exploring this culturally relevant topic that seems to be fairly understudied. However, the reporting and conclusions should be tempered to reflect that this was indeed a preliminary study that should be used as a foundation from which to design future studies.

Throughout the paper, but especially in the introduction, the authors seem concerned that Kratom is having a negative impact on cardiac and autonomic processes (even using the words “cardiotoxicity” and “major concern”). However, the findings are that baseline parasympathetic activity increased in regular Kratom users. This is generally considered a good thing, and is often a marker of successful treatments for stress or anxiety. The authors even state in the intro that “High HRV exhibited in healthy subjects was found to reflect the regulation of the ANS in adaptation to environmental and psychological challenges….”

As a side note, in future studies, it would also be interesting to know in future studies how these autonomic changes correspond to self-report feelings of emotional distress, well-being, sleep quality, etc.

Reviewer 2 ·

Basic reporting

Summary: The current manuscript preliminary explore cardiac autonomic activity with heart rate variability in long-term kratom users.
The authors show that long-term keratom users influenced cardiac autonomic function.
Although the authors present interesting findings some aspects could be improved.

Introduction: Overall, the introduction provides a broad background and rationale for the research. However, needs more specific detail. I suggest that you improve the description at lines 67- 80 (Regarding HRV) to provide more justification for your study (specifically, you should expand the knowledge gap regarding association with HRV).

Methods: The method is concise and comprehensive. However, some gaps generate confusion. For example, because in the tables is named a measure of Very High Frequencies and in the text is not reported its nature. To my knowledge, no study reports this information. Major detail is needed for non-linear analyses.

Results: The study summary is well-defined and fits according to the analysis plan provided. I suggest removing figure 1 and better explaining figure 2.

Discussion: the discussion appears to be a summary of the results. I suggest reporting the usefulness of this study and further perspective. A broader literature should be addressed about HRV. In addition, I think the authors could go further in some personal considerations, trying better to explain the highlighted relationship from a neuropsychology perspective

General comment: I would also encourage the authors to check all references and proofread the manuscript to improve the English language.

Experimental design

no comment

Validity of the findings

no comment

---

## Round 0.2 · accepted · Accept

Confirm that the authors have addressed all of the reviewers' comments.